# Development and Characterization of Eudragit^®^ EPO-Based Solid Dispersion of Rosuvastatin Calcium to Foresee the Impact on Solubility, Dissolution and Antihyperlipidemic Activity

**DOI:** 10.3390/ph15040492

**Published:** 2022-04-18

**Authors:** Sana Inam, Muhammad Irfan, Noor ul ain Lali, Haroon Khalid Syed, Sajid Asghar, Ikram Ullah Khan, Salah-Ud-Din Khan, Muhammad Shahid Iqbal, Imran Zaheer, Ahmed Khames, Heba A. Abou-Taleb, Mohammad A. S. Abourehab

**Affiliations:** 1Department of Pharmaceutics, Faculty of Pharmaceutical Sciences, Government College University, Faisalabad 38000, Pakistan; drsanainam@gmail.com (S.I.); or syedharoonkhalid@gcuf.edu.pk (H.K.S.); sajuhappa@gmail.com (S.A.); ikramglt@gmail.com (I.U.K.); 2Department of Medicine, Fatima Jinnah Medical University, Lahore 42000, Pakistan; lalinoor79@gmail.com; 3Department of Biochemistry, College of Medicine, Imam Mohammad Ibn Saud Islamic University (IMSIU), Riyadh 11432, Saudi Arabia; sdikhan@imamu.edu.sa; 4Department of Clinical Pharmacy, College of Pharmacy, Prince Sattam bin Abdulaziz University, Alkharj 11942, Saudi Arabia; 5Department of Pharmacology, College of Medicine, Al-Dawadmi Campus, Shaqra University, Al-Dawadmi 11961, Saudi Arabia; imran82zaheer@gmail.com; 6Department of Pharmaceutics and Industrial Pharmacy, College of Pharmacy, Taif University, P.O. Box 11099, Taif 21944, Saudi Arabia; a.khamies@tu.edu.sa; 7Department of Pharmaceutics and Industrial Pharmacy, Faculty of Pharmacy, Merit University (MUE), New Sohag 82755, Egypt; heba.ahmed@merit.edu.eg; 8Department of Pharmaceutics, Faculty of Pharmacy, Umm Al-Qura University, Makkah 21955, Saudi Arabia; maabourehab@uqu.edu.sa; 9Department of Pharmaceutics and Industrial Pharmacy, College of Pharmacy, Minia University, Minia 61519, Egypt

**Keywords:** solid dispersion, rosuvastatin calcium, Eudragit^®^ EPO, docking, improved solubility, anti-hyperlipidemic activity

## Abstract

Poor solubility is the major challenge involved in the formulation development of new chemical entities (NCEs), as more than 40% of NCEs are practically insoluble in water. Solid dispersion (SD) is a promising technology for improving dissolution and, thereby, the bioavailability of poorly soluble drugs. This study investigates the influence of a pH-sensitive acrylate polymer, EPO, on the physicochemical properties of rosuvastatin calcium, an antihyperlipidemic drug. In silico docking was conducted with numerous polymers to predict drug polymer miscibility. The screened-out polymer was used to fabricate the binary SD of RoC in variable ratios using the co-grinding and solvent evaporation methods. The prepared formulations were assessed for physiochemical parameters such as saturation solubility, drug content and in vitro drug release. The optimized formulations were further ruled out using solid-state characterization (FTIR, DSC, XRD and SEM) and in vitro cytotoxicity. The results revealed that all SDs profoundly increased solubility as well as drug release. However, the formulation RSE-2, with a remarkable 71.88-fold increase in solubility, presented 92% of drug release in the initial 5 min. The molecular interaction studied using FTIR, XRD, DSC and SEM analysis evidenced the improvement of in vitro dissolution. The enhancement in solubility of RoC may be important for the modulation of the dyslipidemia response. Therefore, pharmacodynamic activity was conducted for optimized formulations. Our findings suggested an ameliorative effect of RSE-2 in dyslipidemia and its associated complications. Moreover, RSE-2 exhibited nonexistence of cytotoxicity against human liver cell lines. Convincingly, this study demonstrates that SD of RoC can be successfully fabricated by EPO, and have all the characteristics that are favourable for superior dissolution and better therapeutic response to the drug.

## 1. Introduction

The oral route has been found to be the most convenient route for taking medications, owing to ease of administration and flexibility in the design of dosage systems [1]. It is frequently documented that nearly 40% of newly discovered drugs have low solubility and reduced bioavailability, regardless of their high permeability [2].This poor solubility of drugs at various pH levels of the gastrointestinal fluid is a gigantic challenge in attaining the desired therapeutic response. Thus, formulation scientists are constantly putting dynamic efforts into developing strategies to advance the dissolution and/or apparent solubility of poorly soluble drugs into orally bioavailable drugs [3].

Numerous formulation schemes—such as liposomes, nanoparticles [4], self-emulsifying drug delivery systems (SMEDDS) [5], hydrotrophy [6], cyclodextrin complexation [7], cosolvency [8], micronization [9,10], chemical modification [11], solid lipid nanoparticles [12] and amorphous solid dispersion (ASD) [13,14,15,16] are utilized in research to address the problems of poorly soluble drugs. Currently, the fast-tracked progression of FDA-approved drugs has decisively made solid dispersion (SD) an established technology for the design of hydrophobic drugs. SD consists of at least two different components: generally, a hydrophilic matrix that is either crystalline or amorphous, and a hydrophobic drug that is dispersed either as a discrete molecule or as an amorphous/crystalline particle [17]. The enhanced dissolution rate of drugs from SDs may be reinforced by the reduction in particle size to a molecular level, improved wettability and increased porosity [18]. SD can be prepared using numerous methods such as fusion [19], precipitation [20], trituration [21], solvent evaporation [22], kneading [23], spray drying [24], hot-melt extrusion [25], lyophilization [20] and electrospinning [26].

Rosuvastatin calcium (RoC) is the most effective antihyperlipidemic drug that competitively inhibits hydroxymethyl glutaryl coenzyme A (HMG-CoA) reductase and decreases the biosynthesis of cholesterol. It is mainly designated for the treatment of dyslipidemia. Unlike other statins, it is considered to be highly potent and well tolerated in humans for the management of hyperlipidemia [27]. However, being a BCS Class II drug, RoC mainly exhibits poor oral bioavailability of 20% due to low aqueous solubility of 0.01 mg/mL [28]. This reduced solubility of RoC influences its dissolution rate and, in turn, its bioavailability. Therefore, there is a need for a system that can augment the dissolution and oral bioavailability of RoC [29].

The appropriate selection of a hydrophilic carrier for SDs has a significant impact on the release of drugs in the upper part of the GIT. Eudragit^®^ EPO (MW: 100,000–250,000 g/mol) is basically a cationic terpolymer based on methyl methacrylate, N-N-dimethyl aminoethyl methacrylate and butyl methacrylate monomers in the ratio of 1:2:1. EPO was selected because the tertiary amine groups that ionize at an acidic pH make it highly soluble up to pH 5. Additionally, it is also swellable and permeable in the GIT above pH 5. These features make it a successful candidate for augmenting the solubility of hydrophobic drugs [30,31,32].

The intent of this study was to use the SD technique to improve the solubility, as well as the dissolution, of BCS Class II rosuvastatin calcium (RoC) in the upper part of the GIT by using pH-dependent methacrylate polymer (EPO). Additionally, the uniqueness of this study lies in the fact the EPO-based SDs were first investigated for the improvement of anti-hyperlipidemic activity. To achieve the goal line, the study was thoroughly characterized and evaluated to meet a set level of improvement in terms of bioavailability and dyslipidemia.

## 2. Results

### 2.1. Saturation Solubility Studies

RoC has low aqueous solubility of 0.051 mg/mL and 0.421 mg/mL in distilled water and 0.1 N HCl, respectively (Figure 1). The study of solubility conducted in bio-relevant media shows a drastic increase of 4054.76, 4059.52, 6531.14, 5942.86, 5647.62, 4080.95, 7100, 6773.80, 5611.90 and 4647.62% for RoC-1, RoC-2, RoC-3, RoC-4, RoC-5, RSE-1, RSE-2, RSE-3, RSE-4, and RSE-5, respectively (*p* < 0.05). Thus, the hydrophilic carrier (EPO) had a significant role in enhancing the solubility of the pure drug (RoC) in the upper part of the GIT. The maximum and significant (*p* < 0.05) solubility is found to be for RoC-3 (27.88 mg/mL) and RSE-2 (32.4 mg/mL) using the CG (co-grinding) and SE (solvent evaporation) methods, respectively.

### 2.2. Gibb’s Free Energy Calculation

Gibb’s free energy is basically an indicator of the spontaneous solubilization behavior of a drug. The values for the Gibbs free energy are negative and in the range of 13,034.50 to 11,309.70. The higher negative Gibbs energy values of −12,385.67 and −12,627.16 are revealed by RoC-3 and RSE-2, respectively.

### 2.3. Percentage Yield and Drug Loading

The prepared formulations were assessed for product yield and drug content (Table 1) to govern the efficiency of the used methods. It is found that all the formulations demonstrated good product yields ranging from 63 ± 0.84 to 96.87 ± 0.18%. The drug content is in the range of 90.01 ± 0.29 to 98.50 ± 0.88 which is within accepted pharmacopeia limits. RSE-2 confines the highest amount (98.50 ± 0.88%) of RoC in contrast to RSE-1 and RoC-5. Based on the saturation solubility, Gibbs energy, percentage yield and percentage drug content, RoC-3 and RSE-2 were selected for further studies, including FTIR, XRD, DSC, SEM and PD.

### 2.4. FTIR Analysis

FTIR analysis is a procedure broadly utilized for the determination of chemical contact between drugs and excipients. The FTIR spectrum of RoC (Figure 2a) exhibits evidence of quite a lot of characteristic peaks at 1546.6 cm^−1^ (aromatic C–N stretching vibrations), 1507.3 cm^−1^ ( C–C stretching in aromatic ring), 1383.6 cm^−1^ (aromatic C–F stretching vibration), 1322.9 cm^−1^ (sulfone asymmetric stretching) and 1223.45 cm^−1^ (C=O stretch in carbonyl group). The intense broad bands at 3339.1 cm^−1^ are due to carboxylic OH stretching and are suggestive of H-bonding between two molecules of RoC.

The FTIR spectrum of EPO (Figure 2b) shows weak hydroxyl (O–H) stretching at 3437.43 cm^−1^, a strong aromatic C–H stretching band at 2954.81cm^−1^, a strong band of alkene C–H stretching at 2814.88 cm^−1^, an intermediate alkene C=C stretching bend at 1449 cm^−1^, a strong bend of C–O stretching of the ester group at 1389.21 cm^−1^ and a strong bend of C–C stretching at 1149.19 cm^−1^. The sharp peak at 1721 cm^−1^ is indicative of the occurrence of carbonyl groups in EPO polymer. Moreover, the characteristic functional group peak for dimethyl amino groups is observed between 2770 and 2824 cm^−1^.

The significant similarities are perceived in the spectra of pure RoC with reference to physical mixture (1:1), RoC-3 and RSE-2 (Figure 2c–e). There is insignificant alteration in the peak pattern of RoC in RoC-3 and RSE-2, which confirms the lack of chemical interaction within the components of SDs.

### 2.5. Thermal Analysis

The thermal behavior and possible physicochemical interaction between the ingredients were further investigated using DSC to foresee the shifting and disappearance of peaks in SDs. A DSC thermogram (Figure 3) of pure RoC shows multiple onsets of glass transition peaks in the ranges of 123–170 °C, and one deep endothermic peak at 219.89 °C corresponding to its fusion and degradation. This predicts the semi-crystalline structure of pure RoC. The pure hydrophilic polymer (EPO) does not show a melting endotherm because of its amorphous nature. The DSC curves of RoC-3 and RSE-2 (also presented in enlarged form in Appendix A) are seen to be flattened, corresponding to the strong molecular dispersion and entrapment of RoC inside the EPO polymeric matrix. The TGA thermograms, Appendix A, showed that onset of decomposition was delayed in both RoC-3 and RSE-2. Moreover, thermal stability was found much better in RSE-2 in contrast to the pure drug (RoC).

### 2.6. PXRD Analysis

The PXRD spectra of pure drug, EPO and SDs (RoC-3 and RSE-2) are presented in Figure 4. The pure drug indicates a semi-crystalline nature, with characteristic peaks at 8.05°, 12.95° and 22.2° (depicted by arrows). Halo patterns are obtained for EPOs with no intrinsic peaks. The complete disappearance of diffractogram peaks, such as 8.05°, 12.95° and 22.2° in the optimized SDs, may be attributed to transformation into an amorphous state with no crystallization during the formulation stages. This confirms that both the drug and the polymer existed in an amorphous state in both RoC-3 and RSE-2.

### 2.7. SEM Analysis

A micrograph of the pure RoC (Figure 5a) depicts well-defined crystals with rectangular dimensions and rough edges, which is quite typical of the semi-crystalline nature. The microphotograph of physical mixture (Figure 5b) revealed differentiation in the manner as smaller particles of EPO were seemed to be adhered to the bigger particles of RoC. The SEM of RoC-3 (Figure 5c) exposed amorphous structure with irregular surface edges whereas the SEM image of RSE-2 (Figure 5d) revealed more asymmetrical and amorphous features that is quite typical of the lack of original shape of RoC.

### 2.8. In Vitro Dissolution Study

The dissolution profiles of pure RoC and SDs in dissolution media maintained under sink conditions are illustrated in Figure 6 and Figure 7. In the initial 5 min, pure RoC exhibits the slowest release rate of only 16%, and more than 40% of the drug is released from each of the SDs (Figure 6). At 60 min, when the dissolution rate of the pure RoC is approximately 48.02%, the release rates for RoC-1, RoC-2, RoC-3, RoC-4 and RoC-5 are 88.78%, 89.74%, 96.91%, 92.99% and 91.19%, respectively. RoC-3 has the highest dissolution rate of 80.03% in the first 5 min (*p* < 0.05) among all of the dissolution profiles using the CG method.

In case of the SE method, dissolution rates are 90.91%, 99.68%, 98.33%, 96.51%, and 93.81% for RSE-1, RSE-2, RSE-3, RSE-4 and RSE-5, respectively, at 60 min (Figure 7). Out of all the formulations, RSE-2 exhibits the maximum percentage of drug release (92%) within the initial 5 min (*p* < 0.05). It is evident that the dissolution rates of all the SDs significantly differ from that of the pure RoC (*p* < 0.05). The dissolution profile and statistical analysis suggest domineeringly favorable effects on the dissolution rate of RoC by RoC-3 and RSE-2, and therefore, were designated for PD studies.

### 2.9. Docking Studies for In Silico Prediction of Solubility

The result (Figure 8) demonstrats that complexes with the lowest binding affinity represent the most stable conformation. The docking gold score (binding affinity) presents interaction in the order of RoC-EPO (−7.82 Kcal·mol^−1^) > RoC-Kollidon (−2.74 Kcal·mol^−1^) > RoC-PEG (−1.41 Kcal·mol^−1^). These negative values show that EPO could effectively interact, owing to H-bonding with RoC, and could have the potential to prevent crystal formation of RoC. Moreover, the hydrophobic interaction reveals the entrenchment of drug RoC in the carrier (EPO). This prediction of a better interaction of EPO with RoC is practically confirmed in this study by its enhanced solubility and improved bioavailability.

### 2.10. Pharmacodynamic Studies

#### 2.10.1. Influence on the Gain of Body Weight and Liver Index

The animals in the NC (normal control) group did not show any significant gain (*p* < 0.05) in body weight during the entire experimental period, as they were fed a normal diet. The most common cause for the induction of obesity with hyperlipidemia in rat models is the consumption of a high-fat diet. As is clearly evident in Figure 9, the consumption of a DRF (diet rich in fat) for 6 weeks led to significant gain (*p* < 0.05) of 230.87%, 208.01%, 204.10% and 204.72% in the body weight of rats in the HC (hyperlipidemia control) groups—Group I, Group II and Group III, respectively—compared to the NC group. After the treatment period of 4 weeks, a total increase of 286.56% was observed in the body weight of the HC group, which was found to be statistically significant (*p* < 0.05) in contrast to the NC group (186.06%). During the end of treatment period, the elevated body weights of Group I, Group II and Group III, respectively, presented drastic declines of 24.44%, 57.09% and 66.50%, in contrast to the HC group. The hyperlipidemia-induced rats treated with RoC-3 (Group II) and RSE-2 (Group III), respectively, attained more significant reductions (*p* < 0.05) of 28.45% and 42.06% in their body weight gain in contrast to those who received pure RoC (Group I). Interestingly, among all the treatment groups, Group III appeared to be much more successful at weight gain reduction.

Likewise, the weight of the liver was significantly increased (*p* < 0.001) in the HC group in contrast to the NC group, as graphed by the liver index (Figure 10). The calculated value of the liver index of the HC group was 4.71 ± 0.45 and is comparable to the 2.71 ± 0.08 NC value of the NC group. In contrast, the hyperlipidemic rats treated with pure RoC and SDs (RoC-3, RSE-2) showed a significant decline (*p* < 0.05) of 24.46%, 31.06% and 39.57%, respectively, in liver index, compared to the HC group. The liver index of RoC-3 and RSE-2 was 3.24 ± 0.07 and 2.84 ± 0.15. Thus, it is confirmed that RSE-2 appears to be more effective than ROC-3 and pure RoC (*p* < 0.05) in controlling the liver weight and index.

#### 2.10.2. Biochemical Analysis of Serum Lipid Levels

The measurement of serum lipid levels (Figure 11) reveals a significant (*p* < 0.001) increase in TC (total cholesterol), TG (triglycerides), LDL-C, and VLDL-C levels (115.75 ± 5.73, 187.15 ± 4.98, 94.82 ± 1.37, 37.43 ± 1.98), and a reduction in the HDL-C level (32.05 ± 1.90) of the HC group in comparison to the NC group (74.75 ± 5.84, 70.27 ± 1.94, 52.22 ± 1.36, 37.43 ± 1.22 and 51.11 ± 1.16), respectively. These disturbed levels of lipids in the sera of rats are indicative of dyslipidemia. The hyperlipidemic rats showed improvement in dyslipidemia after the administration of pure RoC and SDs (RoC-3 and RSE-2), simultaneously with DRF, for 4 weeks. The intergroup comparison revealed that RSE-2 causes significant (*p* < 0.05) improvement in the serum lipid levels of TC, TG, LDL-C, VLDL-C and HDL-C to almost normal values of 76.25 ± 6.70, 70.70 ± 3.90, 52.35 ± 5.25, 14.14 ± 1.78 and 49.10 ± 3.07, respectively, contrary to therapy with pure RoC and RoC-3.

The elevated levels of cholesterol and LDL-C and the decreased value of HDL-C have prominent roles in the progression of atherosclerosis, while the anti-hyperlipidemic drugs have the potential to reduce the atherogenic index (A.I) and cardiovascular complications in diabetics with hypercholesterolemia (Figure 12). All the treated groups (I, II, III) reveal a significant (*p* < 0.001) decline of 46.95%, 56.60% and 65.22%, respectively, in the occurrence of atherosclerosis (Figure 12a) compared to HC group (4.1 ± 0.26). There is an insignificant difference (*p* < 0.05) between the A.I of the NC group (1.29 ± 0.029) and group III (1.43 ± 0.14), while the A.I of groups I (2.19 ± 0.19) and II (1.79 ± 0.07) differ significantly (*p* < 0.05), contrary to the NC group. Likewise, both groups (II & III) show a significant (*p* < 0.05) decline in A.I compared to group I. This improved dyslipidemia suggests that EPO-based SDs have the potential to overcome severities, even those prompted by the simultaneous consumption of DRF.

A TG/HDL-C ratio value greater than 3 is an indicator of insulin resistance (I.R), as well as metabolic syndrome, whereby the risk factor for heart disease, diabetes and stroke increases. The results (Figure 12b) indicate the worst TG/HDL-C ratio in the HC group (5.8 ± 0.25) and, thus, present the lowest I.R. All the treatment groups (I, II, III) exhibit a significant (*p* < 0.05) decline in I.R compared to the HC group. Similarly, both groups (II & III) show a significant (*p* < 0.05) decline in I.R compared to group I. The TG/HDL-C ratio for group III (1.45 ± 0.16) is very much similar (*p* < 0.05) to that of the NC group (1.37 ± 0.04).

#### 2.10.3. Biochemical Analysis of Liver Functions (LFTs)

The main organ for lipid metabolism is the liver, so injury and damage to hepatocytes can be estimated by observing elevated levels of liver enzymes. In the present study, the increased levels of ALT and ALP (Figure 13) in the serum of HC rats were the indicator for liver injury. Treatment with ROC and SDs (RoC-3 & RSE-2) significantly (*p* < 0.05) decreased the elevated liver markers compared to the HC group, while group III (DRF + RSE-2) down regulated the elevated levels of ALT and ALP and revealed an insignificant (*p* < 0.05) difference to the NC group.

#### 2.10.4. Macroscopic and Microscopic Examination of Liver

The liver of the NC group (Figure 14a) appeared reddish brown, which is a sign of a healthy liver. Despite this, the liver of hyperlipidemic rats (HC group) appeared enlarged and discolored (Figure 14b) due to consecutive consumption of DRF for 10 weeks. After the treatment period, the visual appearance of the liver transformed from very pale red to more reddish in the order of group III > group II > group I, which was clear evidence of recovery to a healthy red liver (Figure 14c–e).

The histology of normal liver tissues presented hepatocytes with centrally located nuclei and intact cytoplasmic structures (depicted by arrow a) along with no noticeable steatosis (Figure 15a). On the other hand, certain degrees of hepatosteatosis, illustrated by the displacement of nuclei to eccentric position; some ballooning (indicated by arrow b & c, respectively); and slight activation of Kupffer cells (Kc) (Figure 15b), an indication of progressive non-alcoholic fatty liver disease (NAFLD), were observed in rat models of the HC group. All the treatment groups presented reduction in the area of hepatic steatosis and ballooning degeneration, contrary to the HC group. The liver samples of group I (Figure 15c) displayed moderate hepatic degeneration with less vesicular steatosis and fat accumulation. Likewise, these pathological anomalies induced by DRF were almost shifted towards less eccentric nuclei, along with intact cytoplasm and no vesicular ballooning in the order of group III (Figure 15d) is better than group II (Figure 15e).

### 2.11. Cell Viability Assay

A cell viability assay (Figure 16) showed no significant difference between the cytotoxicity of pure RoCa, RSE-2, and the control (*p* < 0.05), suggesting that RoC, along with the polymer (EPO) used, has excellent viability against the human hep G_2_ cell line. Thus, these finding clearly indicate the non-toxicity and biocompatibility of RSE-2 in the formulation.

## 3. Discussion

A number of trials were conducted to finalize the formulation by varying the quantity of the solvent, the time for solvent removal and its solubility in the presence of varying concentrations of polymer. The selection criteria to obtain an optimized formulation were set based on the percentage of drug content, solubility in bio-relevant media, and in vitro release performance. The suitability of SD systems was suggested by the increasingly negative value of ΔG [33]. The established SDs exposed the intensification in the solubility of RoC by both methods, which might be ascribed by intermolecular H-bonding between the carbonyl group of EPO and the hydroxyl group of RoC. Moreover, this improved solubility is thought to be due to reduced particle size and augmented surface area [34]. It was also witnessed that the concentration of polymers played a vital role in the solubility and drug content. The solubility was increased to twice the amount of polymer, and after that, a reduction in solubility was noticed with increasing amounts of polymer. This behavior exhibited by EPO was also noticed with the SDs of curcumin [35] and valsartan [9] Saturation solubility studies conducted in phosphate buffer (pH 6.8) presented a reduction in solubility with increasing amounts of EPO, which is also evidence of the pH-dependent attribute of EPO (data not shown). The good drug content has been suggested to be due to the higher concentration of EPO in SDs [36]. SDs (RSE-2), carrying acetone as a solvent, have been seen to retain the maximal quantity (98.50 ± 0.88%) of RoC, probably due to the better solubility of RoC in acetone [37,38]. These findings suggest an enhancement in solubility and dissolution that can be more clearly seen in optimized SDs (RoC-3 and RSE-2). A computer simulation predicted the strength of the interaction between the drug and the excipients and directed that binding energy, with a higher negative value being suggestive of a lack of phase separation with better stability of the system. Thus, this superlative negative binding energy of EPO for RoC has the potential to form strong H-bonds [39]. These results of docking were found to be in excellent agreement with FTIR results.

The occurrence of peaks at the same wavenumber in the FTIR spectra confirmed the compatibility between the drug and the excipients, with no chemical interactions. The complete absence of a characteristic peak due to O–H bending at 1435.48 cm^−1^, and the shifting of the characteristic peaks of RoC from 3339.1 cm^−1^ to 3127.13 cm^−1^, confirmed the complete entrapment of RoC inside the EPO polymer [40]. The complete disappearance of the melting point peak of the drug in RoC-3 and RSE-2 is suggestive of diminished crystallinity and superior complexation of the hydrophilic polymer (EPO) with the drug. It was established that the heat of fusion of the semi-crystalline RoC was diminished when they were processed by CG and SE techniques [41]. The presence of amorphous polymer EPO in SDs may be an indicator of the conversion of RoC to an amorphous form. The absence of characteristic diffractions peaks in RoC-3 and RSE-2 confirmed that change in the crystal habitat of RoC might be the reason for the enhanced solubility and dissolution. The outcomes of the PXRD studies were proven to be consistent with the DSC studies [42]. The asymmetrical and irregular appearance of the drug in SEM images indicated that SDs had formed drug-loaded milieus with noticeable reduction in the drug crystallinity. This loss of crystallinity of the pure drug was also supported by the perception that the SDs of RoC with the methacrylate polymer (EPO) boosted its solubility and dissolution [43]. The dissolution of orally administered drugs having poor aqueous solubility is an essential consideration in terms of bioavailability. The dissolution behavior of RoC-loaded SDs exhibited better and immediate release compared to that of the pure drug. This intensification in the release of the drug from SDs may originate from numerous mechanisms such as reduced particle size (depicted for RoC-3 and RSE-2 in Appendix A), enhanced surface area and the conversion of a semi-crystalline form to a more thermodynamically stable amorphous form. The rise in dissolution might also be due to the pH-dependent attribute of the hydrophilic carrier, which immediately dissolves and instantly exposes the drug to dissolution medium in the form of fine particles, as there is no lattice energy to overcome unlike crystalline material [35,37]. 

Anti-hyperlipidemic agents were recommended as a targeted approach for the treatment of patients with NAFLD, as approximately 70% of them have coexisting dyslipidemia [40] Additionally, literature has also reported the superior behavior of RoC over other statins in tailoring the dyslipidemia and its associated complications of diabetic dyslipidemia and coronary heart diseases. The current study also took into consideration the impact of the developed formulation on dyslipidemia and its associated complications. The superiority of anti-hyperlipidemic activity [27] and connected I.R (TG/HDL ratio) and A.I of RoC-3 and RSE-2, unlike the pure drug, was probably due to their reduced size, facilitating their uptake and transport across the fenestrae of the liver sinusoids. The better gross appearance and recovery of the enzyme levels of the liver were suggestive of a lack of hepatic deterioration and hepatoprotection [12]. The underlying justification again relies on the diminished size of the particles [44]. Thus, the better performance of RSE-2 compared to the pure drug and RoC-3, from the standpoint of improved hepatic steatosis, biochemical analysis and liver index, was suggestive of an advancement in physicochemical features of RoC; this might be a reason for its improved therapeutic efficiency. Moreover, RSE-2 was found to be bio-safe against the human liver cell line. This was found to be in accordance with the phospholipid-based self-nanoemulsifying systems of RoC with the Caco_2_ cell line [42]. 

## 4. Materials and Methods

### 4.1. Materials

Rosuvastatin calcium (RoC, 98.5% pure) was generously supplied by Saffron Pharmaceutical Pvt. limited Faisalabad, Pakistan. Eudragit^®^ EPO (Tg = 48 °C) was donated by Evonik industries (Piscataway, NJ, USA). Methyl alcohol and acetone of HPLC grade from Daejung chemicals & metals co., Ltd., Shiheung, South Korea were procured by Musaji Adam & Sons, Karachi, Pakistan. Freshly prepared distilled water was consumed throughout the work. All other solvents, chemicals and reagents used in this study were of analytical grade.

### 4.2. In Silico Docking Studies

Computational docking [45] was used to probe drug and polymer interactions at the molecular level and to predict the impact of the selected polymer at the binding site of target proteins. A 3D structure of the receptor HMG-COA reductase (1HWL) was retrieved from the protein databank (PDB). The grid box size for the receptor, and the exhaustiveness were set to 12. Consequently, the difference in the position and root-mean-square deviations (RMSDs) less than 2 Å were clustered together. The energy minimization (EM), a desirable feature prior to docking, for individual as well as drug–polymer structurewas performed, to have the nethermost energy conformations of the complexes. All other parameters remained at default [46]. Then, docking was conducted using GOLD docking software (Version 5.3.0) for monomer units of PEG, Kollidon and EPO. Finally, Discovery Studio Visualizer ver. 19.1.0 (BIOVIA, San Diego, CA, USA) was used for visualization of the protein–ligand complexes, with the objective of optimizing the positions of the drug in the polymer complex. The polymer with the superlative conformation and minimum binding affinity was designated for preparation and evaluation by the SD system.

### 4.3. Methods for Preparation of SDs

Solid dispersions, in different ratios, were prepared using co grinding (CG) and solvent evaporation (SE) methods, as shown in Table 2.

In CG method, the definite amounts of drug and carrier polymer (EPO) were triturated and blended with a mortar and pestle for a period of 30 min. The resulting formulations were milled, passed through a sieve of mesh size 60, and stored in a desiccator until further investigation. The solvent evaporation (SE) method was also used for the preparation of SDs (Figure 17). The required amount of drug and hydrophilic polymer were accurately weighed and dissolved separately into acetone to have a clear solution. The resultant solutions were poured into a 250 mL round-bottom flask and evaporation was carried out below 50 °C using a rotary evaporator (RE-100 Pro, Scilogex, CT, USA). Then the viscous semisolid residue was dried at room temperature over a period of 24 h. The dried mass was pulverized with the aid of a pestle and mortar and passed through a sieve of mesh size 60 to obtain SDs. All the prepared formulations were stored in an air-tight container in a desiccator until further evaluation [35,47]. 

### 4.4. Saturation Solubility Studies

Saturation solubility was determined by employing a shake-flask method using two different media: distilled water and 0.1 N HCl (pH 1.2). An excess amount of pure drug/SD was added into the 3 mL of media in a 15 mL tube. The tube was vigorously vortexed for 5 min and subjected to shaking for 72 h at 37 ± 0.5 °C in a shaking water bath (SWB 15, Thermo-scientific, Waltham, MA, USA) at a speed of 75 rpm. Afterwards, the mixtures were centrifuged at 6000 rpm for 20 min, filtered through a syringe filter (nylon, 0.45 μm) and assayed using a UV–Vis spectrophotometer (CE-7400S, Cecil, Cambridge, UK) at 240 nm. All solubility measurements were performed in triplicate [48]. 

### 4.5. Gibbs-Free-Energy (ΔG°tr) Analysis

This is the utilization of energy formed after an insoluble drug under goes through phase transformation into soluble form [49].The ΔG°tr values of pure RoC and RoC SDs were calculated using the following equation:$$\Delta G{}^{\circ}\mathrm{TR}=\left\{-2.303\mathrm{RT}\;÷{\mathrm{LogS}}_{o}/S_{s}\right\}$$
where S_o_ is the solubility of pure RoC in water and S_s_ is the solubility of RoC SD systems in water. The R is a universal gas constant with a value of 8.31 J/K mol, and T is the temperature in Kelvin [50,51]. 

### 4.6. Determination of Percentage Yield and Drug Content

All prepared SD powders were weighed, and the yield was calculated in percentage by the following equation [52];
$$\mathrm{Yield}\;\left(\%\right)=\frac{\mathrm{weight}\;\mathrm{of}\;\mathrm{dried}\;\mathrm{solid}\;\mathrm{dispersion}}{\mathrm{weight}\;\mathrm{of}\;\mathrm{pure}\;\mathrm{RoC}+\mathrm{EP}0}\times 100$$

20 mg of the sample was accurately weighed and dissolved in 10 mL of methanol. The absorbance of the solution, filtered through 0.45 μm nylon syringe filters, was recorded at 240 nm using a UV–visible spectrophotometer (CE-7400S Cecil, Cambridge, UK) after dilution with acidic buffer media. The drug content was calculated by following equation [53]:$$\mathrm{Drug}\;\mathrm{content}=\frac{\mathrm{Absorbance}\;\mathrm{of}\;\mathrm{sample}\;\mathrm{containg}\;20\;\mathrm{mg}\;\mathrm{of}\;\mathrm{RoC}\;\mathrm{in}\;\mathrm{formulation}}{\mathrm{Absorbance}\;\mathrm{of}\;20\;\mathrm{mg}\;\mathrm{of}\;\mathrm{pure}\;\mathrm{RoC}}\times 100$$

### 4.7. FTIR for Structural Analysis

FTIR spectra were acquired from Agilent Cary 360 FTIR spectrometer (Agilent scientific Instruments, Santa Clara, CA, USA) in KBr discs over a range of 500 cm^−1^ to 4000 cm^−1^ at a resolution of 2 cm^−1^. Data collected in the transmission mode were analyzed using an Essential FTIR [54], to evaluate the interaction of excipients with the drug.

### 4.8. DSC Analysis

The thermograms of samples were recorded using a thermal analyzer (SDT Q600, V20.9 Build 20, TA instruments, New Castle, DE, USA) by sealing samples in an aluminum pan. Analysis was carried out at a heating rate of 10 °C/min with a temperature range of 25 to 400 °C, under a nitrogen purging rate of 20 mL/min [55]. 

### 4.9. PXRD Analysis

The diffraction patterns of samples were obtained using an X-ray powder diffractometer (D8 Advance, Bruker Corporation, Billerica, MA, USA). A sample (10 mg) was packed in an aluminum sample container to obtain the diffraction pattern, by setting the machine to a voltage of 30 kV, a current of 30 mA, from 5° to 60° and at the angular speed of 2θ/min [56].

### 4.10. Scanning Electron Microscopy (SEM)

The shape and surface morphology of the pure RoC and optimized SDs were examined at various magnifications using a Nova-Nano-450 (FEI, Hillsboro, OR, USA) SEM with the aid of a TLD detector at a voltage of 10 kV [13]. 

### 4.11. In Vitro Drug Release

The dissolution behavior of the pure drug and SDs was determined using USP apparatus II (DT 70, Pharma Test, Hainburg, Germany). Accurately weighed quantities of pure RoC and SDs equivalent to 20 mg were placed in vessels containing 900 mL of freshly prepared 0.1 N HCl. The dissolution studies were performed by keeping the temperature of a water bath at 37 ± 0.5 °C and a stirring rate of 100 rpm. Aliquots of 5 mL with replacements of fresh medium were withdrawn at predefined intervals of 5, 10, 15, 30, 45 and 60 min. Samples were filtered and analyzed for the dissolved drug, in triplicate, at a wavelength of 241 nm using a UV–Vis spectrophotometer [57].

### 4.12. Pharmacodynamics (PD) Study in Rats

#### 4.12.1. Experimental Animals

Thirty male Wistar rats, weighing about 140 ± 20 g, with ages ranging from 4–5 weeks were purchased from the animal house of the Government College University, Faisalabad, Pakistan. The animals were kept in a tidy and well-ventilated room at a controlled temperature of 25 ± 2 °C, with 60 ± 5% relative humidity, over 12 h light/dark cycles. Compulsory arrangements were maintained for free access to food and water for all animals. The animal experiments were conducted according to international and institutional guidelines after approval by the Institutional Review Committee, Government College University Faisalabad (Ref No. GCUF/ERC/2068, Study No. 19669, IRB No. 669, 5 September 2019).

#### 4.12.2. Experimental Protocol

In vivo studies were carried out on five groups each comprised of six rats. The experimental protocol (Table 3) was conducted for ten weeks, split into two periods. The initial period comprised of six weeks and was used for the induction of hyperlipidemia with the diet enriched in fat [58]. At the end of dyslipidemia period, the treatment phase with suspension of pure RoC, RoC-3 and RSE-2 was initiated and continued for four weeks. During this time frame, the pure drug and SDs were administrated daily via gastric lavage at a dose of 20 mg/kg of RoC.

The weights of all animals were noted on a weekly basis for the entire period to predict the effect of the fatty diet, as well as treatment, on the body weights of the animals.

#### 4.12.3. Sample Collection

The animals were kept in a fasting state overnight with water provided ad libitum. Blood samples from sacrificed animal were taken and centrifuged at 6000 rpm for 15 min to separate the serum layer. The serum was then stored at −20 °C until further analysis of the lipid and liver profiles. Additionally, the livers were excised, washed with normal saline, and blotted between layers of filter paper for the removal of surface water. Then, the liver index was estimated from the weight of the liver, and the total body weight using the given equation [59].
$$\mathrm{liver}\;\mathrm{index}\;\left(\%\right)=\frac{\mathrm{weight}\;\mathrm{of}\;\mathrm{liver}\;\left(g\right)}{\mathrm{weight}\;\mathrm{of}\;\mathrm{animal}\;\left(g\right)}\times 100$$

#### 4.12.4. Serum Biochemical Analysis

The serum, prepared from fasted blood samples, was analyzed to detect the concentration of triglycerides (TG), total cholesterol (TC), high-density lipoprotein (HDL-C) cholesterol and low-density lipoprotein (LDL-C) cholesterol by employing enzymatic colorimetric methods using commercially available kits (Giesse diagnostics, Guidonia, RM, Italy). The assay was performed according to the guidelines of the manufacturer using an auto-analyzer (Microlab 300, Merck, NJ, USA). The very low-density lipoprotein (VLDL-C) cholesterol was calculated as VLDL = TC/5 [60]. The TG/HDL-C ratio, a marker of insulin resistance, was also calculated in obese models. Moreover, an Atherogenic index (A.I) to estimate the risk of atherosclerosis was calculated using the following formula, as reported in literature [61].
$$A.I\;=\frac{\mathrm{LDL}-C+\mathrm{VLDL}-C}{\mathrm{HDL}-C}$$

Additionally, the biomarkers for LFTs were determined for levels of alanine transaminase (ALT) and alkaline phosphatase (ALP) in the sera using commercial diagnostic kits (Giesse diagnostics, Guidonia, Rome, Italy) to foresee hepatic protection.

#### 4.12.5. Histopathological Examination

For histological examination, the removed livers were fixed in 10% neutral-buffered formaldehyde (NBF) solution. After fixation, the tissue sections were embedded in a paraffin block, segmented at 5 μm, and stained with hematoxylin and eosin (H&E). The microscopic images were captured via Capta Vision software (ACCU-SCOPE 3000, Commack, NY, USA) for the examination of any pathological changes and/or improvement in the individual liver as a result of HFD and/or formulations.

#### 4.12.6. Statistical Analysis

All data obtained from the six rats were expressed as the mean ± standard deviation of the mean. One-way-AVOVA and two way-ANOVA, followed by post-hoc Tuckey test, were performed for the pairwise comparison of columns using Graphpad prism software ver. 5.01 (Graphpad Software, San Diego, CA, USA). Any differences among the groups were considered statistically significant at *p* values less than 0.05.

### 4.13. In Vitro Cell Viability Assay

The in vitro cytotoxicity was assessed by employing an MTT assay using human liver cell line. The Hep G2 cell line was seeded in 96-well plates and incubated at 37 °C for 48 h before treatment with samples. Then, the prepared cells were incubated with pure RoC suspension, and RSE-5 at 37 °C for 48 h with conservation of 5% CO_2_ and 95% fresh air during the whole period. Afterwards, 10 µL of MTT (5 mg/mL stock) solution was added in each well and incubated for next 4 h at 37 °C. The media were aspirated, followed by the addition of DMSO (150 µL) to solubilize the formazan crystals. Finally, the absorbance was measured at 490 nm using an ELIZA plate reader (Type 357, Thermo Fisher Scientific, Shanghai, China), and cell viability was calculated [62] using the following equation:$$\mathrm{Percentage}\;\mathrm{cell}\;\mathrm{viability}=\frac{\mathrm{Absorbance}\;\mathrm{of}\;\mathrm{treated}\;\mathrm{cell}}{\mathrm{Absorbance}\;\mathrm{of}\;\mathrm{control}\;\mathrm{cell}}\times 100$$

## 5. Conclusions

In this study, we successfully prepared SDs of RoC using the CG and SE method. SE showed more promising results than CG in RoC-based SDs. The most satisfactory results were presented by RSE-2 for enhanced solubility with an asymmetrical surface. The in vitro release profiles indicated an immediate release of RoC from the SDs. Our findings present an enhanced vision of pharmacodynamics studies and prove the ameliorative effect of RSE-2 in dyslipidemia, in contrast to the pure drug. Thus, it is concluded that the prepared EPO-based SD is not only safe for the human hep G_2_ cell line, but also has the potential to improve the solubility of the hydrophobic drug (RoC) and the existence of the associated anti-hyperlipidemic effect.

## Figures and Tables

**Figure 1 pharmaceuticals-15-00492-f001:**
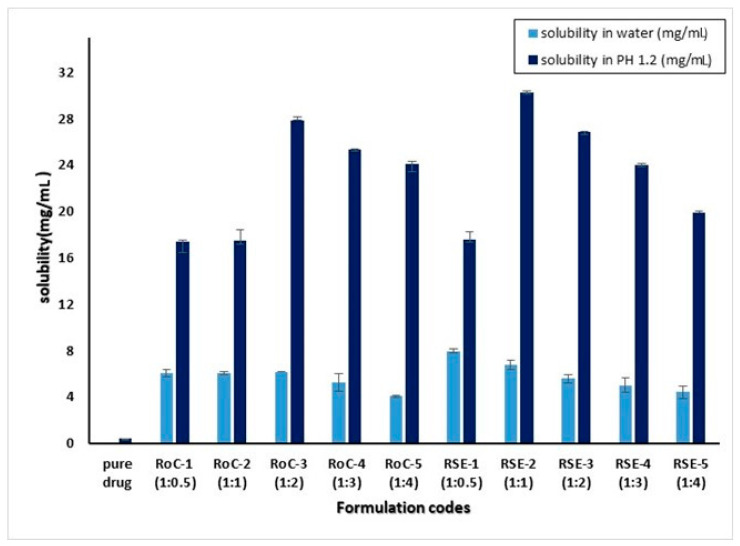
Saturation solubility of pure drug (RoC) and formulations [mean ± S.d; n = 3].

**Figure 2 pharmaceuticals-15-00492-f002:**
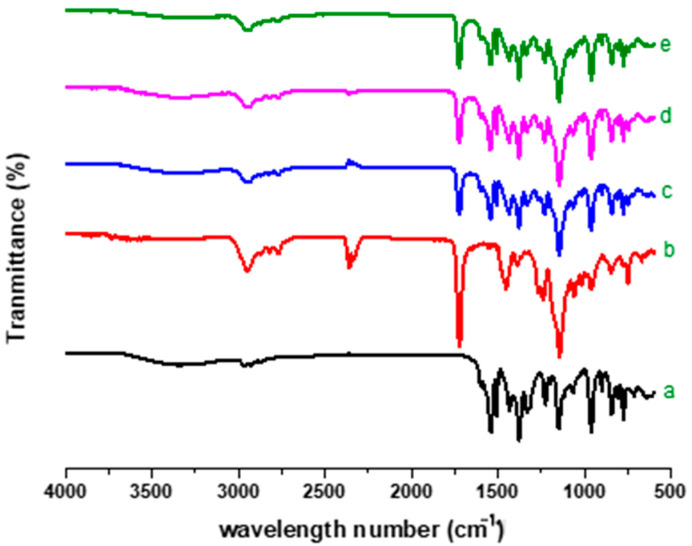
FTIR spectra of (**a**) pure RoC, (**b**) EPO, (**c**) physical mixture, (**d**) RoC-3 and (**e**) RSE-2.

**Figure 3 pharmaceuticals-15-00492-f003:**
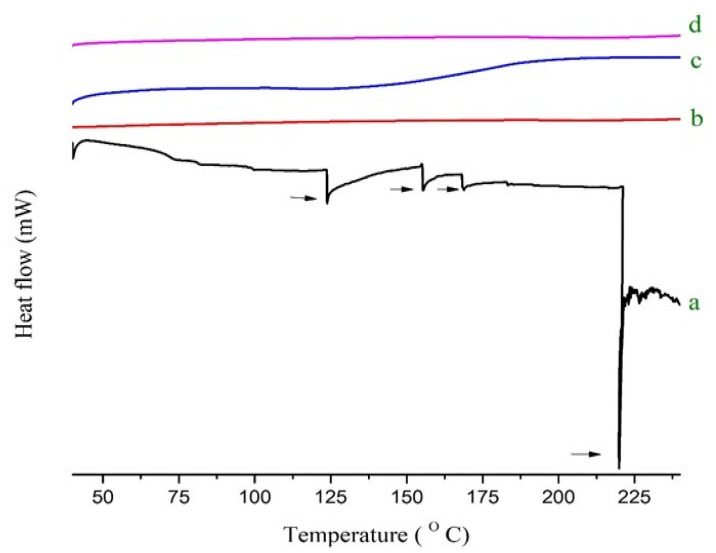
DSC of (**a**) pure RoC, (**b**) EPO, (**c**) RoC-3 and (**d**) RSE-2.

**Figure 4 pharmaceuticals-15-00492-f004:**
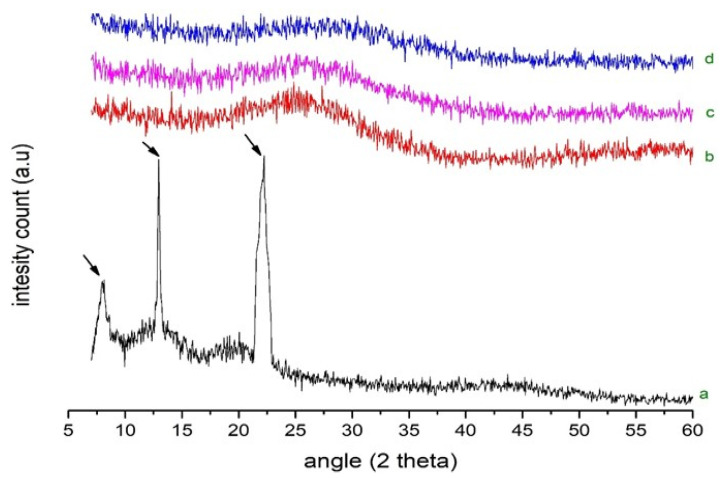
PXRD of (**a**) pure RoC, (**b**) EPO, (**c**) RoC-3 and (**d**) RSE-2.

**Figure 5 pharmaceuticals-15-00492-f005:**
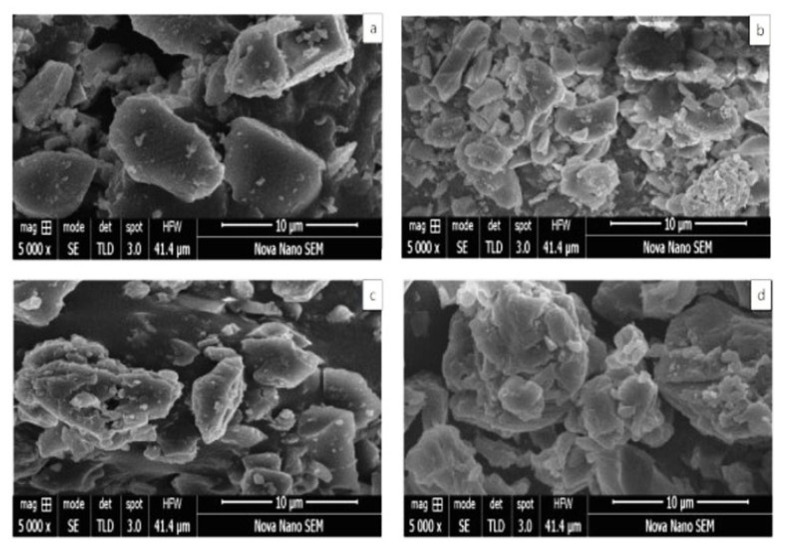
SEM images of (**a**) pure RoC (**b**) physical mixture (**c**) RoC-3 and (**d**) RSE-2.

**Figure 6 pharmaceuticals-15-00492-f006:**
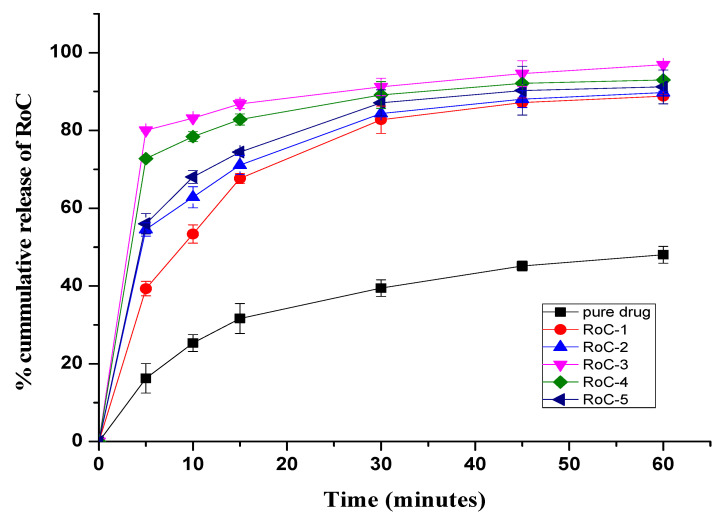
In vitro release profile from pure RoC and SDs prepared using the CG method.

**Figure 7 pharmaceuticals-15-00492-f007:**
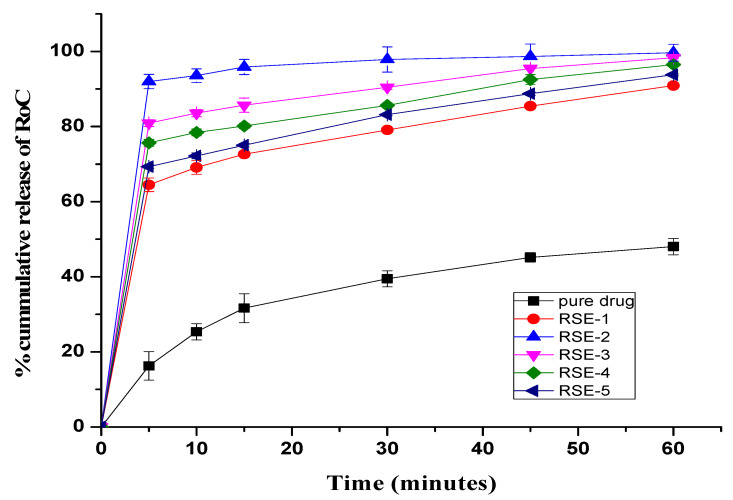
In vitro release profile of pure RoC and SDs prepared using the SE method.

**Figure 8 pharmaceuticals-15-00492-f008:**
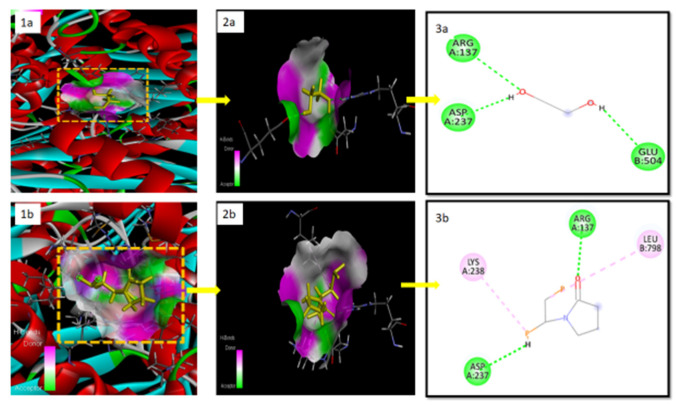
Molecular docking Of RoC with (**a**) PEG, (**b**) Kollidon and (**c**) EPO at the receptor site: 1(**a**–**c**) depict the bonding of monomers shown as yellow colored ligand at the RoC receptor (HMGCOA-reductase); 2(**a**–**c**) present magnified views of 1(**a**–**c**), respectively; and 3(**a**–**c**) depicts amino acids involved in H-bonding. The green dotted lines in 3(**a**–**c**) are symbolic of H-bonding, and pink dotted lines are indicators of hydrophobic interaction.

**Figure 9 pharmaceuticals-15-00492-f009:**
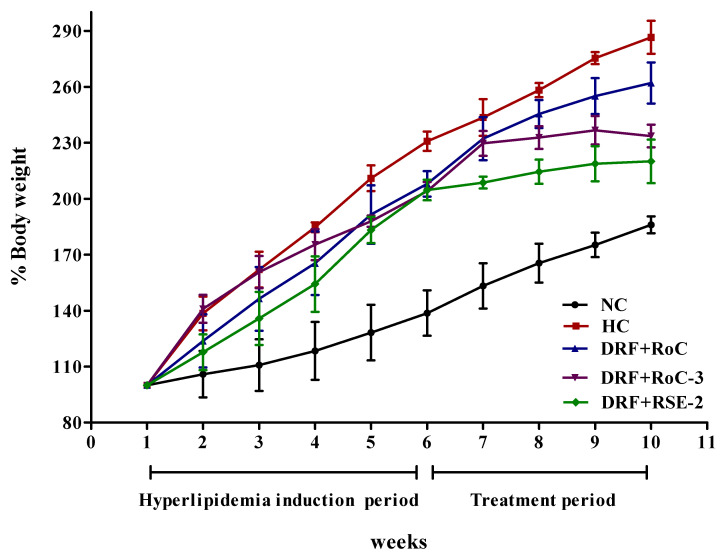
Impact on % body weight of hyperlipidemia-induced rat models (n = 6) after oral administration of DRF (6 weeks) and DRF + RoC-loaded SDs (4 weeks).

**Figure 10 pharmaceuticals-15-00492-f010:**
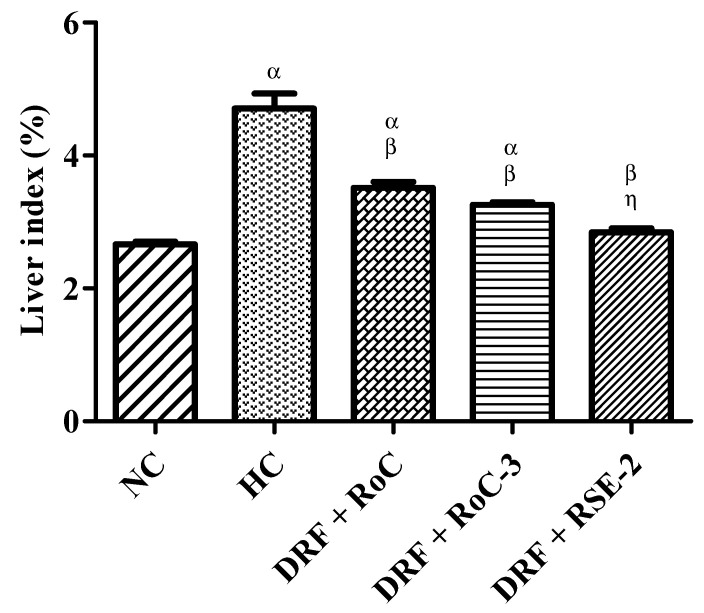
Influence of pure RoC and SDs on the liver index of rats (n = 6) fed DFR: α indicates statistically significant difference (*p* < 0.05) to NC group; β indicates statistically significant difference (*p* < 0.05) to HC group; and η indicates statistically significant difference (*p* < 0.05) to Group I (DRF + Pure RoC).

**Figure 11 pharmaceuticals-15-00492-f011:**
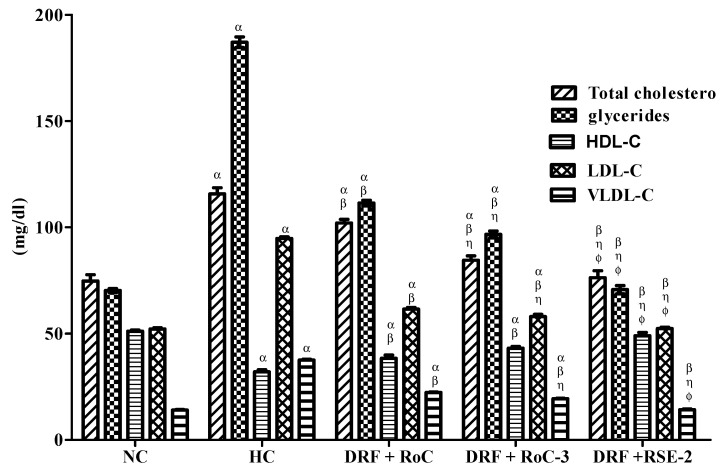
Influence of pure RoC and SDs on the serum lipid levels of rats (n = 6) fed DFR: α indicates statistically significant difference (*p* < 0.05) to NC group; β indicates statistically significant difference (*p* < 0.05) to HC group; η indicates statistically significant difference (*p* < 0.05) to Group I (DRF + Pure RoC); and ϕ indicates statistically significant difference (*p* < 0.05) to Group II (DRF + RoC-3).

**Figure 12 pharmaceuticals-15-00492-f012:**
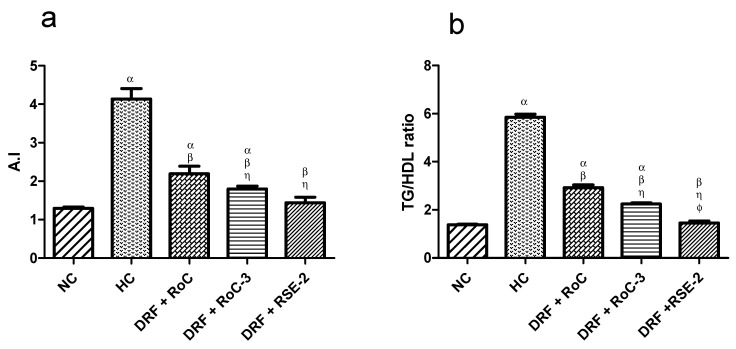
Influence of pure RoC and SDs on the A.I and TG/HDL-C ratio of rats (n = 6) fed on DRF: where α indicates statistically significant difference (*p* < 0.001) to NC group; β indicates statistically significant difference (*p* < 0.05) to HC group; η indicates statistically significant difference (*p* < 0.05) to Group I (DRF + Pure RoC); and ϕ indicates statistically significant difference (*p* < 0.05) to Group II (DRF + RoC-3), (**a**) Effect of pure drug (RoC) and optimized SDs on the Atherogenic index (A.I), (**b**) Effect of pure drug (RoC) and optimized SDs on the TG/HDL-C ratio (an insulin resistance marker).

**Figure 13 pharmaceuticals-15-00492-f013:**
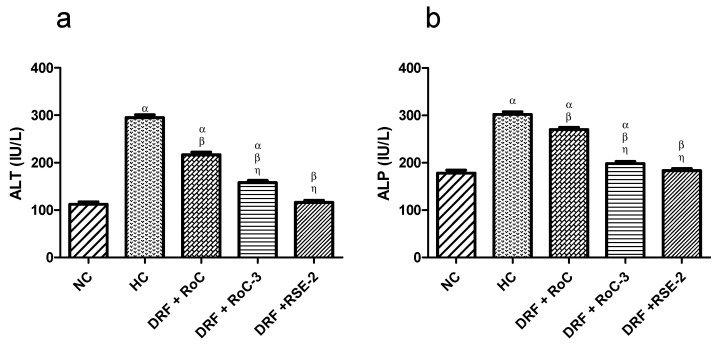
Influence of pure RoC and SDs on the levels of liver enzymes ALT and ALP in rats (n = 6) fed on DRF: where α indicates statistically significant difference (*p* < 0.001) to NC group; β indicates statistically significant difference (*p* < 0.05) to HC group and η indicates statistically significant difference (*p* < 0.05) to Group I (DRF + Pure RoC); (**a**) Effect of pure drug (RoC) and optimized SDs on the levels of liver functioning enzyme alanine transaminase (ALT), (**b**) Effect of pure drug (RoC) and optimized SDs on the levels of liver functioning enzyme alkaline phosphatase (ALP).

**Figure 14 pharmaceuticals-15-00492-f014:**
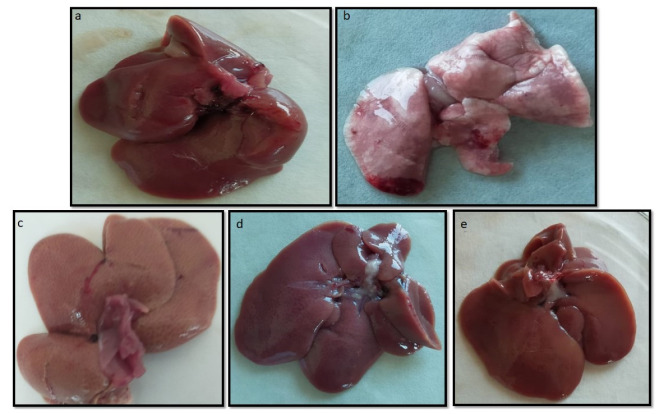
Visual appearance of the liver tissues: (**a**) NC group; (**b**) HC group; (**c**) Group I (DRF + RoC); (**d**) Group II (DRF + RoC-3); and (**e**) Group III (DRF + RSE-2).

**Figure 15 pharmaceuticals-15-00492-f015:**
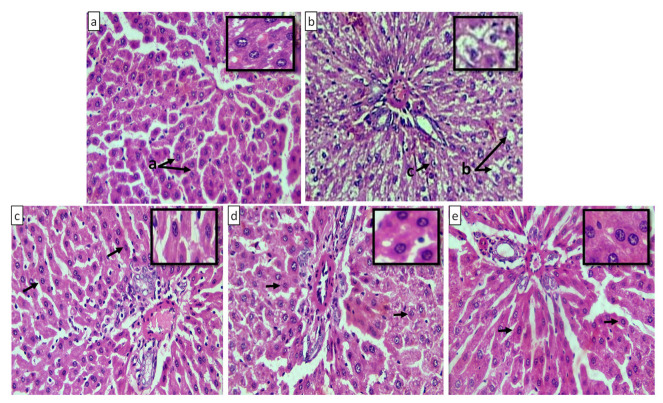
Microscopic images of liver tissues at 40× magnification (H & E staining): (**a**) NC group; (**b**) HC group; (**c**) Group I (DRF + RoC); (**d**) Group II (DRF + RoC-3); and (**e**) Group III (DRF + RSE-2).

**Figure 16 pharmaceuticals-15-00492-f016:**
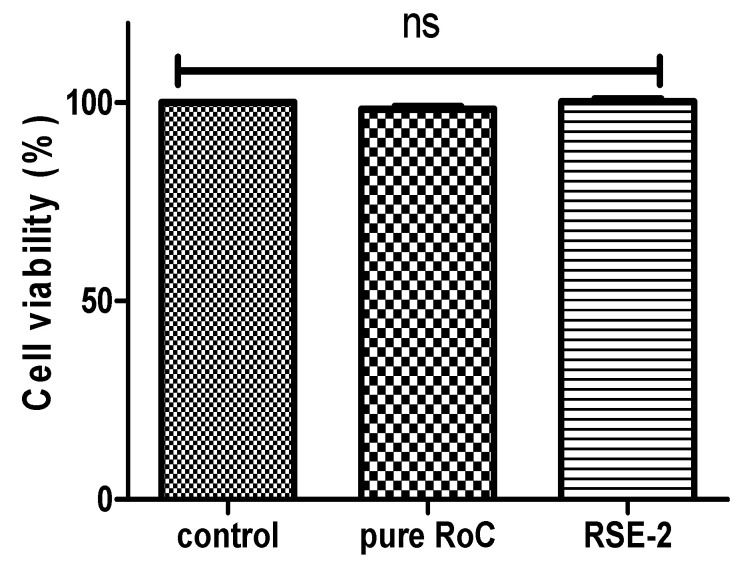
Percent cell viability of control, pure drug and optimized formulation (RSE-2).

**Figure 17 pharmaceuticals-15-00492-f017:**
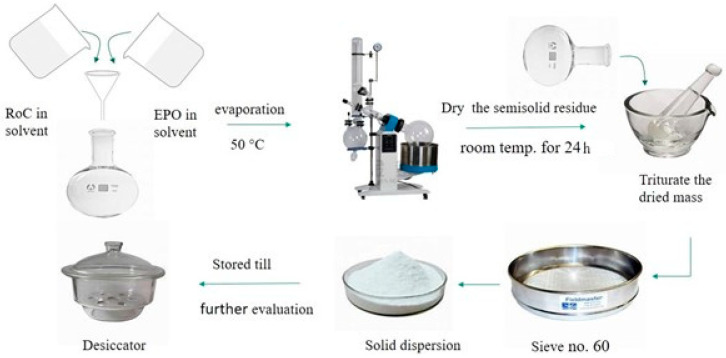
Diagrammatic presentation for the manufacturing process of solid dispersion of RoC using SE technique.

**Table 1 pharmaceuticals-15-00492-t001:** Gibbs energy, percent yield and drug loading of SDs.

Formulation Code	Gibbs Energy	Yield (%)	Drug Content (%)
RoC-1	−12,327.79	90.96 ± 0.42	90.01 ± 0.29
RoC-2	−12,337.81	95.91 ± 0.28	90.69 ± 0.89
RoC-3	−12,385.67	96.87 ± 0.18	91.38 ± 0.83
RoC-4	−11,949.02	95.75 ± 0.21	92.88 ± 1.02
RoC-5	−11,309.71	95.21 ± 0.56	95.48 ± 1.32
RSE-1	−13,034.54	75.73 ± 0.14	96.17 ± 2.13
RSE-2	−12,627.16	79.75 ± 0.35	98.50 ± 0.88
RSE-3	−12,114.25	72.27 ± 0.09	95.89 ± 1.09
RSE-4	−11,846.16	68.75 ± 0.07	96.03 ± 1.04
RSE-5	−11,528.73	63.06 ± 0.84	95.62 ± 1.22

**Table 2 pharmaceuticals-15-00492-t002:** Composition of solid dispersion formulations.

Formulation Code	RoC-1	RoC-2	RoC-3	RoC-4	RoC-5	RSE-1	RSE-2	RSE-3	RSE-4	RSE-5
Ratio of drug	1	1	1	1	1	1	1	1	1	1
Ratio of EPO	0.5	1	2	3	4	0.5	1	2	3	4

**Table 3 pharmaceuticals-15-00492-t003:** Animal grouping for experimental protocol.

Groups	Hyperlipidemia Induction Period (0–6 Weeks)	Treatment Period (7–10 Weeks)
Group NC	SCF	SCF
Group HC	DRF	DFR
Group I	DRF	DFR + pure RoC suspended in 1.33% of CMC solution
Group II	DRF	DRF + RoC-3 (equivalent 20 mg) suspended in 1.33% of CMC solution
Group III	DRF	DRF + RSE-2 (equivalent 20 mg) suspended in 1.33% of CMC solution

NC = normal control; HC = hyperlipidemia control; SCF = standard chow feed; DRF = diet rich in fat (SCF + 35% beef tallow fat); CMC = carboxy methyl cellulose.

## Data Availability

Data is contained within the article and Appendix A.

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
