# Peer review of "Development and Characterization of Eudragit® EPO-Based Solid Dispersion of Rosuvastatin Calcium to Foresee the Impact on Solubility, Dissolution and Antihyperlipidemic Activity"

_pharmaceuticals, 2022, doi:10.3390/ph15040492_

Round 1

Reviewer 1 Report

This manuscript reports the synthesis of several solid dispersions (SD) containing EPO and rosuvastatin. These SDs were prepared by co-milling (CG) and solvent evaporation (SE). Through different analyzes, all these SDs (FT-IR, XRD, DSC, SEM, solubility, dissolution profiles, antihyperlipidemic activities) were evaluated.

Specifically, from the DSC analyses, some doubts arise:

  1. Given the amorphous nature of ROC-3 and RSE-2, can any Tg be distinguished in these two thermograms? This is asked because the five DSC thermograms of Figure 4 must be presented in a more enlarged form in the supplementary material.

2.     The TGA thermograms of samples RoC-3 and RSE-2 should be shown to observe the thermal stability of these two SDs. Since although these thermograms are very flat in Figure 4, and as mentioned, due to the amorphous nature of the samples, it is difficult to determine the melting temperatures, it would be good to see at what temperature the decomposition processes begin.Improve the quality of the nine figures in Figure 9 since they are not well distinguished. Authors are recommended to attend to these observations so that the manuscript can be accepted in this journal.

Author Response

Dear Reviewer 1

Please find the attached reply to your kind suggestions.

Best Regards

Reviewer 2 Report

The paper analyzes the possibilities of improving the bioavailability and performance of solid dispersions based on EPO and rosuvastatin Calcium. It is quite persuasive in showing the fact that solid dispersions perform much better than pure drug. The analysis of whether CG or SE approach and formulation sensitivity analysis are superficial as different samples are only vaguely standardized on the level of microstructure and particle technology. Also, the control choice of pure drug is not as challenging as it would be comparing the dissolution rate to other published rosuvastatin SD systems. More specific comments  follow

  • The article cannot be easily read from the beginning as some terms and abbreviations are defined in the Materials and Methods section, which is placed at the end. If this order of sections should be maintained (I believe it is required by the journal), the result section should be rewritten to contain all necessary definitions while leaving only experimental details for the M&M section.

  • The particle size of dispersions is not adequately characterized. 60 mesh sieve was the only means of particle size control and it is way too large as compared to particle size obtained by grinding. Fig 6 indicate different particle sizes for RoC and RSE formulations. This may affect the interpretation of dissolution data.

  • Solubility results should use physical mixture as control, rather than the pure drug.

  • Saturation solubility measurement protocol uses very short equilibration time which may not reveal the effects of supersaturation and precipitation

  • The USP II data provide only rough information about the release performance due to SD – particle size is nod standardized, USP II is not suitable for powder testing, and again physical mixture should be used as control for comparison rather than the pure drug. How would the authors explain the pure drug release limit to approximately 50 % equilibrium in Figs. 7-8?

  • The initial dissolution rate ratio between RoC and pure drug Fig. 7 is much different from the eq. solubility ratio in fig. 1. Can it be due to precipitation of super saturated solution?

Author Response

Dear Reviewer 2

Please find the attached file having reply to your kind suggestions/questions.

Best Regards
